# Changing Epidemiology of Tinea Capitis in Athens, Greece: The Impact of Immigration and Review of Literature

**DOI:** 10.3390/jof9070703

**Published:** 2023-06-27

**Authors:** Stefanos Charpantidis, Maria Siopi, Georgios Pappas, Kalliopi Theodoridou, Constantinos Tsiamis, George Samonis, Stella-Eugenia Chryssou, Stamatios Gregoriou, Dimitrios Rigopoulos, Athanasios Tsakris, Georgia Vrioni

**Affiliations:** 1Department of Microbiology, “Elena Venizelou” Maternity Hospital, 11521 Athens, Greece; steve_kavala2@hotmail.com; 2Clinical Microbiology Laboratory, “Attikon” University General Hospital, Medical School, National and Kapodistrian University of Athens, 12462 Athens, Greece; marizasiopi@hotmail.com; 3Institute of Continuing Medical Education of Ioannina, 45333 Ioannina, Greece; gpele@otenet.gr; 4Department of Microbiology, Medical School, National and Kapodistrian University of Athens, 11527 Athens, Greece; lmktheo@yahoo.com (K.T.); atsakris@med.uoa.gr (A.T.); 5Department of Microbiology, “Andreas Syggros” Hospital for Skin and Venereal Diseases, 16121 Athens, Greece; syggroslab@gmail.com; 6Department of Public and Integrated Health, School of Health Sciences, University of Thessaly, 43100 Karditsa, Greece; ktsiamis@uth.gr; 7Department of Internal Medicine, School of Medicine, University of Crete, 71003 Heraklion Crete, Greece; samonis@med.uoc.gr; 81st Department of Dermatology and Venereology, “Andreas Syggros” Hospital for Skin and Venereal Diseases, National and Kapodistrian University of Athens, 16121 Athens, Greece; stamgreg@yahoo.gr (S.G.); dimitrisrigopoulos54@gmail.com (D.R.)

**Keywords:** tinea capitis, epidemiology, Greece, immigration, anthropophilic dermatophytes

## Abstract

Mass population movements have altered the epidemiology of tinea capitis (TC) in countries receiving refugees. Periodic monitoring of the local pathogen profiles may serve as a basis for both the selection of appropriate empirical antifungal therapy and the implementation of preventive actions. Therefore, we investigated the impact of an unprecedented immigration wave occurring in Greece since 2015 on the epidemiological trends of TC. All microbiologically confirmed TC cases diagnosed during the period 2012–2019 in a referral academic hospital for dermatological disorders in Athens, Greece, were retrospectively reviewed. A total of 583 patients were recorded, where 348 (60%) were male, 547 (94%) were children and 160 (27%) were immigrants from Balkan, Middle Eastern, Asian as well as African countries. The overall annual incidence of TC was 0.49, with a significant increase over the years (*p* = 0.007). *M. canis* was the predominant causative agent (74%), followed by *T. violaceum* (12%), *T. tonsurans* (7%) and other rare dermatophyte species (7%). *M. canis* prevalence decreased from 2014 to 2019 (84% to 67%, *p* = 0.021) in parallel with a three-fold increase in *T. violaceum* plus *T. tonsurans* rates (10% to 32%, *p* = 0.002). An increasing incidence of TC with a shift towards anthropophilic *Trichophyton* spp. in Greece could be linked to the immigration flows from different socioeconomic backgrounds.

## 1. Introduction

Tinea capitis (TC) remains a predominantly pediatric, superficial scalp infection with remarkable morbidity in endemic populations [1]. Significant progress has been made in minimizing the disease burden in developed countries, partly through the availability of adequate antifungal therapies, most prominently griseofulvin, although it is nowadays not available in certain European countries [2,3]. Yet, worldwide eradication has been far from achieved, despite efforts [1]. Meanwhile, the epidemiology of TC has markedly changed during the last years under the influence of several factors, such as globalization and climate change, that have modified its incidence as well as the pattern of the causative dermatophyte species [1,4,5]. Of note, the epidemiological trends of TC vary widely across different geographic areas, even within a given country [1]. At the same time, the emergence of anthropophilic infections, particularly in European urban areas, which can promote epidemics, raises concern [1]. Hence, periodic and updated assessments of the local epidemiological aspects of TC are crucial since they can serve as a basis for both the selection of appropriate empirical antifungal therapy and the implementation of preventive actions.

The ecology of infectious diseases has traditionally been influenced by large-scale immigration flows, voluntary or forced, among others [6]. During the last decades, forced, political or economic reasons have caused massive immigration to the European countries. Greece has been the recipient of two major immigration waves in recent years; one at the beginning of the 1990s, after the fall of communist regimes in Eastern Europe and the Balkans, and the second, still evolving since 2015, with the entrance of great numbers of Middle Eastern, Asian and African populations, arriving mainly through Turkey [7]. These populations’ movements, with different socioeconomic backgrounds and infectious disease ecology, unavoidably affect the epidemiological trends of skin diseases in the recipient country [4,5]. Such an effect was observed during the first immigration wave in Greece regarding various skin infections, including TC [8]. To date, the contemporary epidemiology of TC in our country remains unknown. Based on these grounds, we (i) performed a systematic review of the existing literature related to TC in Greece and (ii) investigated the impact of the second immigration wave on the epidemiological aspects of TC in Greece by studying cases reported in the largest referral hospital for dermatological disorders in Athens, Greece, during a period of 8 years, starting in 2012 and continuing through 2015 (the initiation of the second immigration wave) until 2019.

## 2. Materials and Methods

### 2.1. Literature Review

We carried out electronic searches in the PubMed database using the keywords “tinea capitis”, “tinea” and “dermatophytes” in conjunction with “Greece” and/or “Greek”, whereas additional handpicked searches in the bibliographies of the articles retrieved were also performed. Only studies written in the English language, without restriction on the year of publication, were considered. From the articles retrieved, we extracted data concerning the geographic region, the study period and the population of each, the number of cases, the morbidity rates as well as the relative proportions of dermatophyte species.

### 2.2. Study Setting

A retrospective study of outpatients with clinically suspected TC attending the “Andreas Syggros” University Hospital in Athens between 1 January 2012 and 31 December 2019 was conducted. Of note, “Andreas Syggros” is a Greek tertiary referral center for dermatological disorders, covering almost half of the national population. Demographic data (gender, age, country of origin) obtained during routine patient visits as well as laboratory findings were reviewed. All data accessed were anonymized and individual patient consent was deemed not required.

TC diagnosis was confirmed by positive direct microscopic examination and/or culture. Skin scrapings and pulled hair from patients with a clinical manifestation of TC, including swollen red patches, dry scaly rashes, severe itchiness and alopecia, were collected by scraping the actively growing edge of the infected area or by using a scalp brush, such as a disposable toothbrush or swab, following an aseptic technique. A mycological investigation was subsequently performed by conventional methods consisting of direct microscopic examination for fungal elements using 10% potassium hydroxide without dimethyl sulfoxide and culture onto appropriate agar plates (Sabouraud dextrose agar and Sabouraud dextrose agar with cycloheximide) incubated at 30 °C for up to 4 weeks. Recovered isolates were identified to the genus and species level by experienced medical microbiologists on the basis of the macromorphology of the colonies (texture, pigmentation of the front/reverse side, rate of growth) and their microscopic morphology. Additional physiological tests were performed for species identification, including the hydrolysis of urea and in vitro hair perforation capacity of the fungus [9].

### 2.3. Statistical Analysis

The annual incidence of TC was calculated by dividing the number of episodes by the study population at risk, while its trends over time were evaluated by linear regression analysis and ANOVA, followed by a post test for linear trend. Medians and interquartile ranges (IQR) were calculated for continuous variables, whereas numbers and percentages were calculated for categorical parameters. Categorical variables were compared by Pearson’s chi-square test. In any case, a two-tailed *p* value of <0.05 was considered to reveal a statistically significant difference. All data were analyzed using the statistics software package GraphPad Prism, version 8.0, for Windows (GraphPad Software, San Diego, CA, USA).

## 3. Results

### 3.1. Literature Review

We retrieved 11 articles regarding the epidemiological aspects of TC in Greece whose extracted data are presented in Table 1. Overall, the study periods encompassed the years 1981 to 2015 (7/11; 64% before 2008). All studies reported data from single hospitals located in Northern Greece (Thessaloniki, 6/11; 54%), Crete (3/11; 28%), Southwestern Greece (Patras, 1/11; 9%) and Athens (1/11; 9%). The majority of them (8/11; 73%) were performed in the general patient population, 2/11 (18%) in pediatric patients and 1/11 (9%) in adult patients. Over the years, the most commonly isolated dermatophyte species was *M. canis*, with a frequency ranging from 73% to 94%, regardless of the geographic area. Among anthropophilic spp., *T. violaceum* was the most frequently isolated (1.1 to 54%), followed by *T. rubrum* (0.4 to 9%), *T. tonsurans* (0.3 to 3%) and *T. soudanense* (0.5 to 1.5%).

### 3.2. Demographic Data and Incidence of the Present Study

During the 8-year study period, 1191 individual patients (289; 24% of non-Greek origin) with clinically suspected TC were recorded, whereof 583 (49%) eventually had a microbiologically confirmed diagnosis (Table 2). Namely, cultures yielded dermatophytes in 540 (93%) cases, whereas 43 (7%) were positive only with direct microscopy. The median (range, IQR) number of TC cases reported per year was 76 (51–97, 30). The overall (range) annual incidence of TC was 0.49 (0.38–0.60), with a significant increase over the years, particularly between 2012–2013, 2014–2015 and 2016–2019 (0.46, 0.41 and 0.56, respectively; *p =* 0.007) (Figure 1).

There were 348 (60%) male and 235 (40%) female patients with TC (*p* = 0.003). Their median (range, IQR) age was 6 (0.5–83, 4) years. In particular, 3 (0.5%) cases occurred in infants, 547 (94%) in children of preschool and school age (89% younger than 10 years old), 9 (1.5%) in adults between 18 and 59 years old and 24 (4%) in elderly patients (≥60 years old). Concerning their ethnic origin, 423 were native and 160 were immigrants from Balkan, Middle Eastern, Asian as well as African countries. In total, positivity rates for a TC diagnosis were significantly higher in non-Greek patients evaluated than in indigenous patients (55% versus 47%, *p* = 0.012) (Table 2).

### 3.3. Distribution of Dermatophyte Species

As regards the spectrum of the etiological factors of TC, the majority of the isolated dermatophytes were zoophilic (76%), followed by anthropophilic (23%) and geophilic (1%). The annual causative agent distribution is illustrated in Figure 1. Overall, *M. canis* was the predominant TC-related pathogen (*n* = 400; 74%), followed by *T. violaceum* (*n* = 66; 12%) and *T. tonsurans* (*n* = 38; 7%). Other dermatophyte species were randomly recognized, sometimes in clusters, as observed with a *T. soudanense* cluster in 2016, with a significant presence in the overall annual epidemiological profile (*n* = 36; 7%). The dispersion of dermatophyte species isolates did not differ according to the patient’s gender (*p* = 0.36). *M. canis* was the major pathogen in native patients (85% versus 45%; *p* < 0.0001), while *T. tonsurans* and *T. violaceum* predominated in the immigrant population (5% versus 14%; *p* = 0.0003 and 5% versus 30%; *p* < 0.0001, respectively) (Table 2).

In total, the number of cultures yielding *M. canis* was significantly higher (*p* < 0.0001) than those yielding *T. tonsurans* and *T. violaceum* during the immigration wave era from 2015 through 2019. Nonetheless, the isolation rate of *M. canis* significantly decreased from 84% in 2014 to 67% in 2019 (*p* = 0.021). In parallel, the prevalence of *T. tonsurans* and *T. violaceum* infections significantly increased from 3% in 2014 to 14% in 2019 (*p* = 0.032) and from 7% in 2014 to 18% in 2019 (*p* = 0.047), respectively (Figure 1). Overall, the sum of positive *T. tonsurans* plus *T. violaceum* cultures presented a three-fold increase in incidence during the years 2014–2019, i.e., from 10% in 2014 to 32% in 2019 (*p* = 0.002). Due to the random isolation of other dermatophyte species, no statistical results could be extracted from the study records.

## 4. Discussion

Global population mobility is increasingly linked to the evolving epidemiological landscape of TC worldwide. The current study outlines the effect of an unprecedented immigration wave in Greece, progressing since 2015, on the epidemiological trends of TC. During the 8-year (2012–2019) period, a significant rise in the incidence of the infection was observed. A notable decline in *M. canis* rates, still dominant though, in parallel with a three-fold increased isolation of anthropophilic dermatophyte species, with a predilection for immigrants, has been recorded.

Consistently with most other reports, we observed that males had a significantly higher predisposition to TC (male-to-female ratio of 1.5) [1,20,21,22]. Male preponderance could be related to their short hair, which facilitates the implantation of spores and their transmission, as well as the need for frequent trimming of the hair, probably by contaminated scissors and blades [1]. In addition, the age distribution of our patients suffering from TC revealed the well-described epidemiology of the disease, with the majority of them (89%) being pre-adolescent children. Actually, the lack of sebum excretion during the prepubescent period results in decreased medium chain length fatty acids and increased pH on the scalp that, in turn, promote dermatophyte colonization and subsequent infection [2]. Furthermore, the ethnic origin of the patients was associated with the emergence of the infection, as the proportion of immigrants with TC was significantly higher compared to the native population (*p* = 0.012). In fact, refugee and immigrant communities are commonly of a low socioeconomic status and are vulnerable to adverse health conditions since they reside in dense living conditions characterized by a lack of proper hygiene, paving the way for the spread of tinea infections.

The etiology of TC has been traditionally associated with socioeconomic factors, mainly poverty and destitution. The history of the disease in Greece is a typical example. In the 1960s, social and financial population parameters were different between the rapidly developing and expanding urban centers and the still underdeveloped rural areas, and this schism was also obvious in the etiology of TC, with the majority of cases countrywide attributed to *T. violaceum* (77%) and a lower percentage (20%) caused by *M. canis*. All cases of *M. canis,* though, originated in urban centers (Athens, Thessaloniki, Chania, Heraklion, Patras), while almost 95% of the *T. violaceum* strains originated in islands and villages [23]. It is noteworthy that *T. schoenleinii* being the cause of the more severe clinical entity called “favus”, at that time, was absent in the present series [23].

Subsequent epidemiological studies demonstrated the persistence of *T. violaceum* as the major causative agent of TC in adults in Northern Greece in the period 1981–1995, whereas *M. canis* predominated in children of the same area, underlining the general improvement of the country’s hygienic situation and the ensuing correlation of TC with zoonotic transmission from pets [12]. Further reports from the same region in the following years (2004–2014) demonstrated the almost exclusive presence of *M. canis* as an etiological agent of TC in children [17]. *M. canis* was implicated in pediatric cases from other geographic areas of the country as well [16], with an expanding role of this pathogen in children suffering from TC on the island of Crete, whereas non-*M. canis*-induced cases in elderly patients were extremely scarce, as shown by serial epidemiological studies of all dermatophytoses in this area [15,19]. Regarding the evaluation of the Athens samples from the second half of the ‘90s onward (after the peak of the first immigration wave), anthropophilic fungi had already become significantly apparent (in 12% of the cases, predominantly in immigrants) [8]. Another study focusing on *T. violaceum* specifically outlined its re-emergence due to the influx of immigrants, mainly from neighboring Albania [24]. A surveillance study investigating household contacts of patients with TC, at the same time, demonstrated the significance of *T. violaceum* in these settings [25].

Greece was the frontline of Europe’s migration crisis at the beginning of 2015, when a massive inflow of refugees, mainly from Syria, Iraq, Afghanistan and some sub-Saharan African areas, landed in our country [7]. Our findings provide a picture of the current epidemiology of TC following this still-evolving demographic event. Interestingly, the incidence of the infection increased over the years (*p* = 0.007), accompanied by a gradual decrease in *M. canis* rates (84% in 2014 to 67% in 2019, *p* = 0.021) concomitantly with a rise in *T. violaceum* plus *T. tonsurans* rates (10% in 2014 to 32% in 2019, *p* = 0.002). In fact, the marked increase in the incidence of TC may be ascribed to the influx of immigrants originating from countries where the disease is endemic as well the shift in the pathogenic spectrum of TC since the transmission of anthropophilic dermatophytoses can even occur through indirect contact with patient-contaminated belongings or environments, and thus facilitates the spread of the infection to others.

The migration of dermatophytes owing to the population movements as a result of socio-political and economic conflicts has been observed worldwide, corroborating our findings. The replacement of *M. canis* by *T. violaceum* and *M. audouinii* as the dominant pathogens of TC in Tel Aviv could be linked to a considerable increase in refugees from Eastern Africa who migrated to Israel [26]. Similarly, the remarkable rise in scalp infections due to *M. audouinii* and *T. soudanense* in North-Western Spain, Belgium as well as the Montreal area (the third largest host of immigrant populations in Canada) might reflect the increased immigration from African countries [21,27,28]. Moreover, our data are consistent with several studies demonstrating that *M. canis* was still the main causative agent of TC in European native populations, as opposed to immigrants where anthropophilic dermatophyte species predominated [29,30]. Actually, the increased isolation rate of anthropophilic strains with a predilection for immigrants is unsurprising since hygienic conditions in detention facilities are far from satisfactory, allowing clusters of TC cases in children [1].

Of note, the tendency of a pathogenic shift towards anthropophilic dermatophyte species giving rise to TC in our region was apparent already after the peak of the first immigration wave. Frangoulis et al. reported *T. violaceum* as the second most frequent pathogen of TC during the period 1996–2001, accounting for 8% of cases, followed by other anthropophilic fungi (*M. audouinii*, *M. ferrugineum*, *T. soudanense* and *T. schoenleinii*; 4% of cases) that were isolated for the first time only in infected immigrant children, while *T. tonsurans*-induced infections were lacking [8]. Our data reaffirm this epidemiological trend revealing, however, higher *T. violaceum* rates (12%) in conjunction with a doubling incidence of anthropophilic dermatophyte species isolation (23% of cases), whereof *T. tonsurans* accounted for 7% of scalp infections. A possible explanation is the changing profiles of immigration waves in our country during the last decades, as previously stated. Notably, *T. tonsurans* has been constantly recognized as an emerging cause of TC in numerous industrialized countries [20,31,32,33,34,35], yet it remains a random causative agent in the Greek population (4%); whether the rise in isolation rate observed in 2019 (14% versus 3% in 2014) denotes a generally increasing trend, remains to be further evaluated in the following years.

A limitation of the present study comprises the lack of detailed individual patient data (risk factors for TC, antifungal therapy, clinical outcome) given its retrospective nature and the discontinuation of follow-up evaluations after only one visit for the vast majority of cases. Nevertheless, the study fills a gap in the existing literature and provides an updated view of the epidemiology of TC in our country over an 8-year period, which can be of vital importance in designing local therapeutic strategies and preventive actions. Certainly, it should be further noted that although the present survey was carried out in a tertiary referral hospital located in Athens, accounting for the examination of a significant percentage of immigrants, our findings may underestimate the overall effect of immigration on the etiology of TC nationwide. Tens of thousands of refugees are currently residing in camps and shelters in Eastern Aegean islands or other peripheral locations in Greece [36], making it possible that active TC cases are going undetected on a larger scale and consequently undertreated due to inadequate access of immigrants to specialized dermatological centers. Moreover, the phenotypic identification of dermatophytes is sometimes difficult or uncertain because there are variations from one isolate to another and even overlapping characteristics between species. The combination of conventional and molecular-based methods could strongly add value to the accurate diagnosis of fungal infections, and thus the lack of molecular identification of dermatophytes could be considered a potential limitation of our study. However, the implementation of various molecular identification procedures in laboratory routine is of little practical use except in mid-to-high-level reference laboratories due to their resource requirements and equipment. Of note, during the study period (2012–2019), an unprecedented financial crisis affected our country, leading to limited resources for medical care, difficulties in diagnosis and treatment as well as reduced numbers of specialized hospital staff [37,38]. In fact, limited PCR testing capacity has been recently reported by hospital-based microbiological laboratories both locally [39] and globally [40,41,42,43,44], while having the capacity does not translate into routine testing due to a lack of funding.

## 5. Conclusions

The current study provides a contemporary overview of the epidemiological trends of TC in Greece since 2012. The incidence of TC increased significantly during the last years and a shift towards anthropophilic *Trichophyton* spp. was observed, reflecting the impact of mass population movements on the ecology of a mycotic scalp infection with great contagious potential. This up-to-date knowledge about the etiological factors of TC circulating locally may facilitate the microbiological differential diagnosis and the surveillance of social and family contacts as well. Additionally, reporting the local experience may also influence the selection of empirical therapy since many specialists suggest the use of terbinafine as the first choice of treatment for *Trichophyton*-induced infections [45].

## Figures and Tables

**Figure 1 jof-09-00703-f001:**
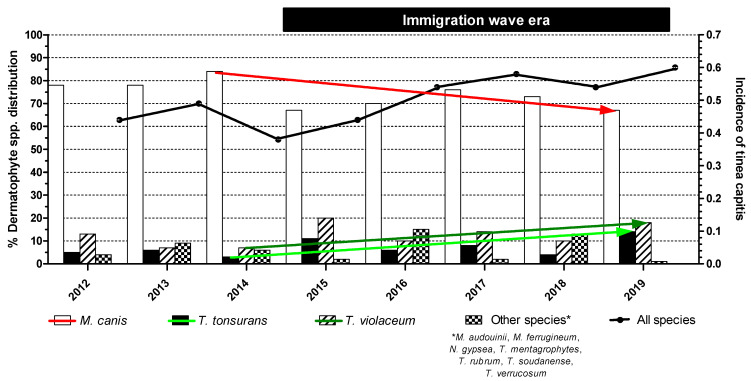
Species distribution of dermatophyte isolates and temporal changes in tinea capitis episodes. Statistically significant differences between the isolation rates of *M. canis* (red falling arrow, *p* = 0.021), *T. tonsurans* (light green rising arrow, *p* = 0.032) and *T. violaceum* (light green rising arrow, *p* = 0.047) as well as in the incidence of the infection (black line, *p* = 0.007) were recorded during the immigration wave era.

**Table 1 jof-09-00703-t001:** Epidemiology of tinea capitis (TC) in Greece as outlined in various studies presented in chronological order.

Author [Ref.]	Observation Time Period	Study Design	Study Population	Number of Cases	TC Incidence	Dermatophyte Species
Devliotou-Panagiotidou et al. [10]	1981–1990	Retrospective, single-center (Thessaloniki)	General patient population	314	4.8% of all dermatophytoses	*M. canis* (80%), *T. violaceum* (12%), *T. mentagrophytes* (3%), *T. verrucosum* (2%), *T. schoenleinii* (1%), *N. gypsea* (0.6%), *T. rubrum* (0.6%), *T. tonsurans* (0.6%)
Koussidou-Eremondi et al. [11]	1981–1995	Retrospective, single-center (Thessaloniki)	Pediatric patients	559	Not available	*M. canis* (88%), *T. violaceum* (6%), *T. mentagrophytes* (2%), *N. gypsea* (1.5%), *T. verrucosum* (1%), *T. schoenleinii* (0.7%), *T. soudanense* (0.5%), *T. tonsurans* (0.3%)
Devliotou-Panagiotidou et al. [12]	1981–1995	Retrospective, single-center (Thessaloniki)	Adult patients	35	Not available	*T. violaceum* (54%), *M. canis* (14%), *T. rubrum* (9%), *T. verrucosum* (9%), *T. mentagrophytes* (5.5%), *T. schoenleinii* (5.5%), *T. tonsurans* (3%)
Maraki et al. [13]	1992–1996	Retrospective, single-center (Crete)	General patient population	37	11% of all dermatophytoses	*M. canis* (73%), *T. violaceum* (16%), *T. rubrum* (8%), *N. gypsea* (3%)
Koussidou-Eremondi et al. [14]	1996–2000	Retrospective, single-center (Thessaloniki)	Pediatric patients	280	46% of all dermatophytoses	*M. canis* (98.5%), *T. violaceum* (1.1%), *N. gypsea* (0.4%)
Frangoulis et al. [8]	1996–2001	Retrospective, single-center (Athens)	General patient population	577	46% of clinically suspected TC	*M. canis* (84.5%), *T. violaceum* (8.4%), *M. audouinii* (1.6%), *T. mentagrophytes* (1.5%), *M. ferrugineum* (1.3%), *N. gypsea* (1.1%), *T. soudanense* (0.5%), *T. rubrum* (0.4%), *T. verrucosum* (0.4%), *T. erinacei* (0.2%), *T. schoenleinii* (0.2%)
Maraki et al. [15]	1997–2003	Retrospective, single-center (Crete)	General patient population	61	12% of all dermatophytoses	*M. canis* (80%), *T. violaceum* (7%), *N. gypsea* (5%), *T. mentagrophytes* (3.5%), *T. soudanense* (1.5%), *T. verrucosum* (1.5%), *Trichophyton* spp. (1.5%)
Tsoumani et al. [16]	1991–2008	Retrospective, single-center (Patras)	General patient population	91	12% of all dermatophytoses	*M. canis* (77%), *T. mentagrophytes* (16%), *T. rubrum* (6%), *N. gypsea* (1%)
Chokoeva et al. [17]	2004–2014	Retrospective, single-center (Thessaloniki)	General patient population	253	2.5% of all dermatophytoses (27% and 0.1% in the pediatric and adult population, respectively)	*M. canis* (92%), *T. violaceum* (3%), *T. tonsurans* (1%), *T. mentagrophytes* (2%), *T. verrucosum* (2%)
Nasr et al. [18]	2010–2014	Retrospective, single-center (Thessaloniki)	General patient population	9	50% of clinically suspected TC	*M. canis* (89%), Not available (11%)
Maraki et al. [19]	2011–2015	Retrospective, single-center (Crete)	General patient population	35	12% of all dermatophytoses	*M. canis* (94%), *T. mentagrophytes* (6%)

**Table 2 jof-09-00703-t002:** Distribution of tinea capitis (TC) cases and prevalence of dermatophyte species by country of origin.

Population		Greek	Immigrant
Number of patients with clinically suspected TC		902	289
Number (%) of patients with microbiologically confirmed TC		**423 (47%)**	**160 (55%)**
Number of patients with culture-positive TC diagnosis		388	152
TC etiological factor			
	*M. audouinii*	1 (0.3%)	0 (0%)
	*M. canis*	**331 (85%)**	**69 (45%)**
	*M. ferrugineum*	0 (0%)	1 (0.7%)
	*N. gypsea*	3 (1%)	1 (0.7%)
	*T. mentagrophytes*	9 (2%)	1 (0.7%)
	*T. rubrum*	4 (1%)	0 (0%)
	*T. soudanense*	**1 (0.3%)**	**14 (9%)**
	*T. tonsurans*	**17 (5%)**	**21 (14%)**
	*T. verrucosum*	1 (0.3%)	0 (0%)
	*T. violaceum*	**21 (5%)**	**45 (30%)**

Bold values indicate a statistically significant difference at the *p* < 0.05 level.

## Data Availability

Data available on request.

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
