# Peer review of "Changing Epidemiology of Tinea Capitis in Athens, Greece: The Impact of Immigration and Review of Literature"

_jof, 2023, doi:10.3390/jof9070703_

Round 1

Reviewer 1 Report

The study presents new epidemiological data on tinea capitis from tertiary hospital in Athens during the 8-year period. Changing epidemiology with a shift to anthropophilic dermatophytes reflects the influence of immigration wave in Greece.

Comments:

-        The study presents data from a single tertiary hospital in Athens, therefore, the results can not be interpreted and are not representative for the Greece in general. Therefore, the title of the article should specify more precisely the geographic region, as was also defined in the previous articles on tinea capitis in Greece.

-          Results of this study are compared with previous studies, published from other geographic regions. The authors should explain if the results could also be compared with previous data from the same region.

-          Data on clinical characteristics of tinea capitis are unfortunately missing.

-        Identification of dermatophytes based only on the culture and microscopic examination may be difficult in some patients. Were additional agars or PCR used for identification of dermatophyte in difficult cases?

The paper is well written from linguistic point of view.

Author Response

Please find the attach file

Reviewer 2 Report

Dear Authors,

in my opinion your work is very interesting in a cognitive context and contributes a lot to medical mycology, epidemiology of dermatophytoses and also in the context of socioeconomic studies. The research conducted by you are very important to show the impact of immigration and globalization on changing epidemiology of dermatophytoses.

All the tables and figures are appropriate for this type of article. In general, the paper has a logical flow. The abstract well correspond with the main aspects of the work and the literature is well selected for well prepared "Introduction" and "Discussion". Nevertheless, the main weakness of this work may be the lack of support of the results obtained based on conventional mycological diagnostics by molecular biology techniques. It would be ideal and it would certainly increase the value of this work if species identification of the obtained dermatophytes were performed based on genomic DNA isolation, PCR amplification of the ITS region and sequencing of the obtained amplicons. The specific nucleotide sequences of the amplicons obtained by you could then be deposited and available to those interested in the GenBank or other database. This type of approach in fungal species identification combining traditional/conventional and molecular mycological diagnostics is complete and is becoming a standard in microbiological laboratories around the world.

As a reviewer I am obligated to pay attention even to less important weak points of this work and all mentioned below comments should be carefully considered. All the comments below do not diminish my high assessment of this work.

Abstract

Lines 25-26

In my opinion ,, ... implementation of preventive actions...” or ,, ... implementation of preventive procedures ...” sounds better

Lines 29-30

I would suggest a slight correction to this sentence, namely ,,A total of 583 patients were qualified, where 348 (60%) were...”

Line 33

Instead of "rare dermatophyte spp." should be "rare dermatophyte species" As I know such abbreviation ,,spp.” is usually used in conjunction with the generic name of the fungus e.g. Trichophyton spp., Microsporum spp., etc., what means that we think about various species of the genus Trichophyton, Microsporum, etc. In my humble opinion, it would be good to improve this throughout the work.

Lines 34 and 35

Instead of "rates decreased from..." I would suggest "prevalence decreased from..." In my opinion ,,rates” should be replaced by ,,prevalence”

Introduction

Line 44

In my humble opinion, in the context of this sentence, it is worth adding that in many countries griseofulvin has been withdrawn (not available) and that more and more attention is being paid to the need to restore its use in therapy in these countries. To the best of my knowledge, griseofulvin is nowadays not available in certain European countries (e.g., Poland, Belgium, Greece, Portugal, and Turkey), let's check (doi: 10.1111/j.1525-1470.2010.01137.x.).

Line 54

,,preventive actions” or ,,preventive procedures” sounds more appropriately than ,,preventive measures”

Line 80

,,morbidity rates” sounds more appropriately than ,,morbidity rates”. Morbidity and mortality are two terms that are commonly used in epidemiology.

Line 90

,,... positive direct microscopic examination...” is more appropriate

Lines 90-92

To the best of my knowledge should be ,,Skin scrapings and pulled hair from patients with a clinical manifestation of TC, including swollen red patches, dry scaly rashes, severe itchiness and alopecia, were collected by ...”

Line 95-96

,,...direct microscopic examination for the presence of fungal elements...” is more adequate

Line 96

Please specify, namely ,,10% potassium hydroxide” reagent included dimethyl sulfoxide (DMSO) or not

Line 99

,,on the basis of macromorphology of colony...” sounds better

Line 107

As I know, there should be space between ,,>” and ,,0.05”

Results

Line 126, Table 1

To the best of my knowledge and based on literature (DOI: 10.1007/s11046-016-0073-9, and also according to Mycobank: https://www.mycobank.org/page/Simple%20names%20search), according to the newest taxonomy of dermatophytes Microsporum gypseum now is called Nannizzia gypsea. Please correct in the text throughout the work.

Line 133

In the context of this sentence (quote) ,,...whereas 43 (7%) were positive only with direct microscopy.” Maybe it's worth adding/explain in the discussion whether it was established on the basis of the interview with the patient whether he/she was treated with antifungal drugs. This could fully explain why cultures were negative while direct microscopic examination was positive.

Line 146

,,Distribution of dermatophyte species” sounds really better

Line 147

,,As regards to the spectrum of etiological factors of TC...” sounds more appropriately.

Line 169, Table 2

Instead of ,,TC pathogen” could be ,,TC etiological factor”. M. gypseum – currently known as Nannizzia gypsea (also within Figure 1).

Line 172, Figure 1

There is an error in the species name. As I know should be T. mentagrophytes.

Discussion

Line 253

To the best of my knowledge should be ,,Of note, the tendency of an epidemiological shift towards anthropophilic dermatophyte...”

Line 274

,,preventive actions” or ,,preventive procedures” is more appropriate in this case

Conclusions

Line 288

,,This up-to-date knowledge about etiological factors of TC circulating locally...” sounds better.

Author Response

Please find the attach file
